# Is Sacred Nature Gendered or Queer? Insights from a Study on Eco-Spiritual Activism in Switzerland

Irene Becci * and Alexandre Grandjean

Institute for Social Sciences of Religions, University of Lausanne, 1015 Lausanne, Switzerland;
alexandre.grandjean@unil.ch
* Correspondence: irene.becci@unil.ch

**Abstract:** Among eco-spiritual activists in the French-speaking part of Switzerland, gendered notions such as "Mother Earth" or gendered "nature spirits" are ubiquitous. Drawing on an in-depth ethnographic study of this milieu (2015–2020), this article presents some of the ways in which these activists articulate gender issues with reference to nature. The authors discuss the centrality of the notion of the self and ask what outputs emerge from linking environmental with spiritual action. We demonstrate that activists in three milieus—the New Age and holistic milieu, the transition network, and neo-shamanism—handle this link differently and thereby give birth to a variety of emic perspectives upon the nature/culture divide, as well as upon gender—ranging from essentialist and organicist views to queer approaches. The authors also present more recent observations on the increasing visibility of women and feminists as key public speakers. They conclude with the importance of contextualizing imaginaries that circulate as universalistic and planetary and of relating them to individuals' gendered selves and their social, political, and economic capital.

**Keywords:** eco-spirituality; holistic practices; New Age; eco-psychology; neo-shamanism; Switzerland; ecofeminism

## 1. Introduction

The scholarly literature on contemporary spirituality has often highlighted that beyond the large variety of practices, beliefs, and views that can be observed, there is a core set of ideas that is shared. As early as the 1990s, Paul Heelas (1996, p. 22), for instance, studied the centrality of the "self" within New Age practices and Françoise Champion (1995) pointed to the primary importance that is given to personal experience, in particular to bodily felt emotions, and optimism in what she termed the "mystico-esoteric nebulae". Moreover, some notions such as "cosmic" or "healing energies"[1] are particularly used in the realm of spirituality that refers to Earth or to nature as a sacred entity. It is in this realm that what is termed "eco-spirituality" emerged; that is, a set of ideas, discourses, and practices articulating religious and environmental considerations in reference to authors calling for a holistic understanding of societies, the human mind, and their environment, such as Ernst Haeckel, Rudolf Steiner, Arne Naess, and Gregory Bateson (Choné 2016). A considerable number of studies have concentrated on the tendency of eco-spiritual initiatives to dichotomize into male and female and essentialize this distinction (Rountree 1999). In this article, we shall contribute to these debates and the literature by offering an analysis of observations made between 2015 and 2020 among Swiss eco-spiritual activists, concentrating on those who commonly use notions such as "Mother Earth" (Grasseli Meier 2018) or "nature spirits" (Chautems and Bressoud 2013; Di Marco and Cruz 2018). We deepen some reflections we have elaborated on previously regarding the way essentialist ideas were reproduced despite subversive claims by eco-spiritual activists (Becci et al. 2020) in the public space. We also rely on more recent findings by authors who have documented emerging "queer spiritual places" in this context (Browne et al. 2010; Lepage 2018; Pike 2001). Their analysis brings nuanced insights into the way gender is performed or subverted by practitioners of contemporary spirituality. These can

subvert conventional gender categories, roles, and boundaries, as well as reproduce them in many ways. Along with existing studies on contemporary spirituality and ecological activism, the present article elaborates on the multiple articulations observed in the field between three core notions: the self, nature, and gender. In order to go beyond the simplistic accusations that often concern ecofeminist views such as essentializing biological and cultural attributes, the analysis is here extended to a large variety of empirically found actors who consider one or the other of the three notions sacred. We shall see that there is a link between a gendered view of a sacred nature and of one's self. To clarify our point, we shall firstly introduce our methodology and the empirical material used. The article then proposes to map the different postures ranging from essentialism to queer approaches according to three milieus found within our fieldwork that are not neatly separated: (1) the New Age and holistic milieu; (2) the transition network; (3) neo-shamanism. It then offers some insights into empirical hubs and contexts where these actors actually meet with feminist key speakers, giving rise to new popular ecofeminist expressions. In the concluding discussion, the importance of contextualizing imaginaries that circulate as universalistic and planetary will be stressed and a link established with individuals' gendered selves and their social, political, and economic capital.

## 2. The Gendered Dimension of the "Spiritualization of Ecology" in Switzerland

Since 2015, in the context of the COP21 Summit in Paris and the diffusion of the *Laudato Sì* encyclical by Pope Francis, a high number of conferences, public demonstrations, urban festivals, and cultural productions in Switzerland have started to contain new forms of eco-activism that include spiritual aspects. Among their different inspirational sources, we found frequent references to neo-shamanism, neo-paganism, green theologies, and holistic approaches. Emotional, subjective, and sensitive ways of being environmentally active have also received growing public attention. We identify this process as a "spiritualization of ecology" (Becci and Grandjean 2021), which is a notion that comes close to what Hubert Knoblauch has termed "the popularization of religion"; that is:

> Religious symbols and forms of religious communication that belonged predominantly or exclusively to the "sacred" religious sphere have been disseminated into other cultural spheres and used in non-religious cultural contexts, most importantly in popular commercial media and leisure culture. (Knoblauch 2008, p. 147)

Through this process, certain symbols, metaphors, or practices become disaffiliated from specific institutions and sometimes become so strongly integrated into an existing cultural setting that they appear as "banal" (Griera and Clot-Garrel 2015). In the case of a spiritualization of ecology, this term refers to the popularization of motives, practices, and symbols (previously present in a narrow circle of environmentalists inspired by esoteric, oriental, or ancient traditions) within a larger semantic and practical field of environmentalism. Interestingly, these motives, practices, and symbols imply gender attributes and references linked to a sacralized Nature.

During public events organized within the milieu of Swiss ecological activists, we observed, between 2015 and 2020, that numerous key speakers articulated religious and spiritual issues with calls for sustainable and societal transformations. By placing emphasis on what they call "spirituality", aiming to, as they state, reconnect with nature and adopt a new posture of articulating the practice of meditation with militancy (Monnot and Grandjean 2021), these key speakers offered discursive and practical tools for individuals to transform themselves in order to change the world.[2] They consider nature and local biotopes at times sacred, harmonious, or even motherly/caring and opposed to humanity, which is presented as being responsible for the current collapse in the Anthropocene. At the beginning of our ethnographic observations of this eco-spiritual milieu, we found that although these approaches joyfully embraced a gendered language, by referring, for instance, to feminine and masculine values, there was no feminist emancipatory agenda in this eco-spiritual sub-milieu (Becci and Grandjean 2018).

During the subsequent four years, we continued our ethnography and have since observed new trends and social innovations appearing in this realm. We therefore complement our initial research conclusions with an analysis of more recent observations that show that militant and affirmative claims about feminism have now arisen in this milieu. The variety of eco-spiritual activism we have observed seems to span from essentialist to queer approaches, all putting forward emic views on what is termed "the feminine" or "the masculine". In order to offer a broader understanding of this oscillation, we refer to the writings of the English philosopher Timothy Morton. Morton has extensively commented upon what he calls "ecological thought" (Morton 2012, p. 19); a perspective that considers all dimensions of life as being ecologically entangled. In that manner, we shall summarize our informants' views and postures about nature, knowing that it is a "promiscuous concept used daily in a multitude of situations by a diverse array of individuals, groups, and organizations" (Castree and Braun 2001, p. 5), paying particular attention to how these are gendered. We also refer to Timothy Morton's reflections on a queer ecology (Morton 2010), providing a non-essentialist and non-binary approach to ecology, also observed in the field. Morton argues that, at a philosophical level, there cannot be any entity such as "Nature" (with a capital "N" to stress its exterior and constructed character) separated from other elements of life (Morton 2010, p. 16). The representation of an allegorical and fantasy Nature is, according to Morton, anchored in Romanticism. Taking up his idea that the more we know about a subject, the less we are actually able to develop a simplistic understanding of it as a discernible whole, we consider that such a view of Nature is eminently metaphorical and rarely tangible. In our discussion, we aim to ground this philosophical insight in our sociological data. We consider that the way the entity called "nature" is gendered or queered mirrors worldviews held within society and social milieus.

### 3. Methodology

We carried out two successive research projects on eco-spiritual activism in Romandie (the French-speaking part of Switzerland) between 2015 and 2020—more precisely in the Lemanic Region (the region surrounding Lake Geneva). We interviewed or had ethnographic conversations with over 70 key eco-spiritual activists in agriculture, holistic spiritualities, environmental advocacy, or Christian Church organizations and observed over 50 public eco-spiritual events. We observed that due to the social vivacity of the area, the social "stocks of knowledge" (Schütz et al. 1974) and practices regarding gender, ecology, and contemporary forms of spirituality that circulate are mixed up with and integrated into each other after having been partly challenged or contested. Our first outline of the concept of a "spiritualization of ecology" arose out of these multisited expressions, together forming a set of ecological references, ideas, and framings that intertwine, gaining public visibility and audibility because they were present in many different milieus, one echoing the other.

In this article, we focus particularly on those public key figures and events entangling spirituality, ecology and gender in visible ways—across books, conferences, workshops, theater plays, street mobilizations, exhibitions, concerts. We rely on the triangulation of the observations and the biographical insights of 18 key public figures, which we can roughly subdivide into three types of spiritual eco-activism: holistic practitioners close to a New Age spirituality, the transition or eco-spiritual network, and neo-shamanic practitioners. A common feature of their sociological profiles is that they have privileged socio-cultural capital and academic diplomas. Their ages range from early thirties to retirement age (over 65 years). Moreover, they often practice liberal or caretaking professions. Most of our informants are also authors of numerous books and other publications, some academic, some belonging to the literature on self-development, eco-spirituality, eco-psychology, Christian contemplation, creative rituality, sacred places, etc. They practice eco-psychological coaching and New Age rituals related to the earth or neo-shamanic rituals.[3] Interestingly, the majority of these key figures are men. At first sight, this finding is surprising as the literature indicates that contemporary spirituality is a milieu where women are most active and predominant (Sointu and Woodhead 2008). It is less surprising, however, if we

consider that at the public events we attended, the speakers were invited as experts on ecological issues, and that in the realm of scientific expertise, a gender gap is still at work and women are strongly underrepresented (Schwaiger et al. 2021). Introducing spiritual views into scientific expertise during public communication is, moreover, an aspect that might diminish scholarly legitimacy, a risk that established senior men take more easily.

As we argue in the following, however, since the year 2017, women have become increasingly present in these milieus as key speakers in public workshops and at conferences. Their increased presence comes in the wake of ecofeminism and goddess spirituality being raised in a number of crucial texts translated into French during the intervening years.[4] New practices and interests such as rituals revolving around menstruation are also more commonly found in these milieus, in mainstream cultural productions, as well as in feminist organizations. The concern over gender issues, addressed in critical or ritualized guises, is yet another recent stance in the eco-spiritual milieu that we have studied.

In addition to semi-structured interviews, we conducted observations at publicly accessible conferences, round-table events, urban festivals, workshops, and spiritual guided tours in parks where these key figures participated. In our ethnographic fieldwork, we paid attention to the broad environment the key figures had access to; that is, we took into account the realm of cultural production, training and education days, workshops, conferences, festivals, etc., they were involved in and documented through notes, photos, and recordings what we observed.

**4. Theoretical Considerations: The Entanglement between Gender, Nature and the Self**

We adopt a constructivist perspective upon what counts as spirituality and religion (Beckford 2003), considering emic discourses and practices as well as controversies arising out of their social definitions. We specifically consider that what is often referred to as spirituality in Western societies is part of a language about "expressive selfhood" and an "attempt to reconcile individuality with relationships in a way that can do justice to both" (Sointu and Woodhead 2008, p. 267). In a similar manner, we analytically consider nature as a polysemic and controversial notion whose study needs empirical grounding in the discourses and practices of social actors. Taking Philippe Descola's considerations about Western ontologies into account, within our analysis we try to avoid sharply dicotomising between what is named nature and what is named culture (Descola 2005). We follow a more phenomenological view, such as that elaborated by Tim Ingold. In particular, we empirically translate his idea that nature and the environment also rely on a "sentient ecology", "consisting of the skills, sensitivities and orientations that humans develop through a life-long experience in a particular environment" (Ingold 2000, p. 24). If the category of nature is a cultural construct, it is also an experiential and relational approach to biotopes, which humans learn to pay attention to and integrate into their local cosmologies. In our pluralistic societies, these cosmologies are made of secular and religious cultures that interlace. Therefore, there is not one eco-spirituality but a multitude of ways to understand and practice the link between what is considered spirituality and what is considered ecology or nature.

Following Linda Woodhead, we consider gender as a "constitutive element of social relations based on the perceived differences between the sexes" and as a "complex and interwoven set of historically constructed relations of domination" (Woodhead 2012, p. 31). Furthermore, following Marylin Strathern, we consider that the common anthropological dichotomy of male and female "provides models of boundaries and relationships" (Strathern 2016, p. 330). Firstly, in relation to such gender boundaries, social actors define something called "the feminine" in stark contrast, sometimes in a complementary way, to what stands as the masculine. Secondly, when objects, symbols and social actors are attributed a specific gender, they are set in relational models and are given ontological limitations: for instance, gendering the planet Earth by calling it "Mother Earth" usually places emphasis on its reproductive and nurturing value while excluding its "dark" (Morton 2018), uncanny, and dreadful dimensions. The notion of "queer" entered scholarly debate through

Teresa De Lauretis's (1991) proposal to reclaim a derogatory term that initially referred to people not falling into one of the two categories proposed by a binary vision of gender. This notion provided an impetus, to scholarly and militant milieus alike, to go beyond a binary categorization and deconstruct the notion of gender in order to avoid confining criticism to the opposition between men–women on the one hand and gays–lesbians on the other. In the realm of ecological critics, "queer" has nonetheless added to the multiplicity of voices, be they human or not, from the Global South or not, that are stuck and are victims, in a non-normative and anti-essentialist manner, of what is often referred to as the Anthropocene (Bauman 2015; Tola 2018). We shall see to what extent the term "queer" does this when it appears in the field of ecological discourses we have observed and analyzed, especially regarding seemingly essentialist and constructivist stances.

## 5. Varieties of Gendered Eco-Spirituality: From Gender Polarity to Queer

In the public spaces of the two main cities where our enquiry took place, Lausanne and Geneva, the presence and circulation of iconographies, discourses, and slogans entangling references to gender, ecology, and spirituality are overwhelming. Roughly every week a new book, public event, or invitation to what is called a "women's circle" or a "holistic treatment" mentioning masculine or feminine energies in relation to nature or the environment is published or circulated. Some of the ecologically committed actors within the milieu of contemporary spirituality were part of the New Age trend, although often stressing the particularity of their own practices. Others explicitly referred to neo-shamanism, while a large number of persons belonged to a loose so-called "transition network" articulating gender issues with spirituality and ecology. We shall discuss which gender views arise within these three eco-spiritual milieus.

### 5.1. The New Age and Holistic Milieu: Gender Complementarity in Nature

We met a number of holistic ecologists during the urban eco-festivals taking place in Lausanne and Geneva over several years (Becci and Okoekpen 2021). We then followed some of them in their activities that drew on gender notions and conducted longer interviews. When analyzing their life-course narratives, we noticed that they had entered the "holistic milieu" (Sointu and Woodhead 2008) smoothly, often after having worked for years in the realm of diplomacy, banking, or in social or international organizations, toward which they had adopted a critical stance and ended up quitting. Quite early on, they had also distanced themselves from the religious traditions they had loosely been socialized in. They have thus passed from a situation of a "distanced" to an "alternative" (Stolz et al. 2016) religious profile in their life course. After having spent some years "seeking" (Sutcliffe 2000, p. 19) different religious expressions, they shared an understanding, as Sutcliffe writes, of "spirituality [ . . . ] as an expressive and holistic mode of personal behavior encouraging strategic interaction on the part of the practitioner between natural and supernatural realms". The "fuzzy boundaries and malleable praxis" this notion of spirituality entails allow them to "occupy ambiguous, multivalent ground between realms elsewhere more clearly categorisable (and hence potentially open to stigma) as 'religious' and 'secular'"(Sutcliffe 2000, p. 20), for instance in the therapeutic or musical realm. They first started their own spiritual practice and, with increasing ecological awareness, they added more explicitly environmental thoughts and metaphors to their practices and references. The figure of nature became important in assessing what they called an "authentic self" as well as in elaborating on gendered and bodily analogies between the planet, animals or plants, and human beings.

In line with most of the literature on the New Age subculture, which insists on an emphasis put on "a higher self" and "on spiritual experience" (Hanegraaff 2002, p. 259), these practitioners are "encouraged to draw inspiration and guidance from within their own minds and bodies rather than from external texts, traditions, or human authorities" (Hedges and Beckford 2000, p. 172). Often considered the achievement of the individualism of modern societies, New Age features the self in its "symbolic center" (Heelas 1996, p. 22).

In the particular context of ecological activism, however, the "authentic self" so constituted would reveal itself in interaction with the environment, "as naturally [ ... ] attuned to the rhythms of the natural world" (Hedges and Beckford 2000, p. 172). Thereby, if this ecologically-oriented holism has a universalizing tendency, it is meant to "foster a sense of compassion for others, rather than individualistic self-improvement" (ibid.).

To illustrate how gender comes into play in this context, we take the example of a Swiss group active in spirituality rooted in geomantic ideas. The members regularly organize visits to what they call "sacred places" to allow people to "reconnect with themselves through the spirits of nature" (Di Marco and Cruz 2018, p. 10). In their book *Nature Spirits in the Parks*, Steeve Di Marco and Gorka Cruz describe the various spirits they have sensed in a great number of public parks in the Lemanic Region. In a Geneva park, they have identified, among numerous other spirits, the presence of Isis, the Egyptian goddess, which they describe as follows: "Isis produces an oscillation in the region of the belly. It opens the access to the deep femininity present in us. She allows us to feel the rhythms of the nature that surrounds us." (Di Marco and Cruz 2018, p. 46) In another nearby park, there is, according to them, a "ram man" described as follows: "This ram-man invites us to tilt our heads forward: the eyes focus on a goal. A will to achieve a goal emerges. Our mind is focused and we cannot be disturbed by what is happening around us." (Di Marco and Cruz 2018, p. 48).

When we participated in one of the visits to the spirits in this park on a Sunday afternoon, the guide showed us how to position ourselves and to be attentive to the energies we felt; that is, we had to stop controlling our bodies and watch what shape they would spontaneously take. He clearly exhorted our little group not to think but to listen to our sensations. This way we could experience the energies that were present and were traversing our bodies. In order to identify what type of spirit it was, he suggested attributes. He asked us whether we were sensing a feminine or a masculine energy. Some participants would go along with these lines, while others would not really follow these categories, sometimes even answering that they felt both or felt androgynous energies. The guide would then explain that a feminine energy was airy, light, and bright and that a masculine energy was strong, focused, and rooted. In an interview, one of the guides told us how he identified these spirits:

> I scan a place, I will feel I will connect, ah, I will feel the feminine vibration and I will say to myself, ah, it is perhaps a fairy who is there. Ah, there it is surely an elf considering that it is straight. (Fieldwork interview, Lausanne, March 2019)

These practitioners consider that nature spirits are present at precise positions all over the world and fulfill specific functions there; some are "cosmic", others are "telluric" (Di Marco 2017). They take care of daisies, they support the growth of trees, they watch and care for the natural order and its balance. For them, humans can enter into contact with these spirits, and the latter can move in time and space if needed. In this cosmology, a spiritualized nature easily entangles with gendered considerations about the self, either through the adoption of binary and stereotypical attributes or in a more nuanced and subtle way. While feminine or masculine energies are not individualized (i.e., attributed to individual persons), the association of the categories with specific contents still reproduces essentialized notions of gender. What characterizes this view most is that the spirits are not standing in tension or power relations to each other. The feminine and masculine energies constitute a polarity without being hierarchically ordered. The spirits are rather active against human drifts such as a so-called "disenchantment of the world" led by Western modernity with its cold rationalism and colonialism. In our informants' views, if humans become tuned to the Nature spirits, they can operate a counter-cultural return to authentic and enchanted selves through spiritual motives. Such ideas appeal to a larger public that participates regularly on the tours and visits without necessarily adopting the whole cosmology. Gender complementarity and nature harmony remain at the core of this view.

*5.2. The Transition Network: The Neutralization of Gender through Organic Metaphors*

We also observed a tendency toward spiritualizing an ecological "self" among practitioners of the transition movement, which calls for an "inner transition", "voluntary sobriety" and economic degrowth. Composed of mostly institutionally established actors, these practitioners operate as ecologists and environmentalists in the first place. They rely on institutional scientific validation and a high level of education (often PhD and professorial level) and are recognized as experts in the realm of ecology. Positions in academia, politics, Christian organizations, and NGOs fostered these key speakers by also providing them with a title offering institutional resources and public legitimacy. Concerning their religious upbringing, these informants have in common a primary Christian socialization, but they had distanced themselves from religious belonging in their early adulthood. The distance from institutional religion resulted in particular from the experience of biographical ruptures, such as divorce or illness. Our informants also shared a grounding of what they called their current "spiritual rediscovery" in something they said had already been present in their childhood, such as nature experiences or specific encounters or trips to Asia or North America. When talking about nature, they delineate symbolic and operative boundaries to distinguish what is not natural but artificial or cultural.

This group gathers around several emblematic authors, such as eco-activist and eco-spiritualist Joana Macy, who has theorized rituals and group exercises known as the "Work that Reconnects" (Macy and Brown 2014). According to this author, who is close to Californian figures and radical ecologists like Starhawk, John Seeds, and Gary Snyder, the current state of ecological and social crisis is leading individuals into a state of despair, helplessness, and anxiety about the future, which the techniques of the "Work that Reconnects" are meant to remedy. In a similar fashion to the holistic milieu, this rhetoric calls for a re-enchantment of the whole world:

> Now, in our time, three rivers, anguish for our world, scientific breakthroughs and ancestral teachings, flow together. From the confluence of these rivers we drink. We awaken to what we once knew: we are alive in a living earth, the source of all we are and can achieve. Despite our conditioning by the industrial society of the last two centuries, we want to name, once again, this world as holy. (Macy and Brown 2014, p. 14)

These authors propose an eco-psychology explicitly interlacing references to exotic ancestry and to new modes of human relations to nature conceived as sacred or holy. Because of these references, the terms "eco-psychology" and "eco-spirituality" are used in an interchangeable way depending on the audience. Their approach stands on the fringes of the modernist distinction between what counts as religious and what counts as secular, thus promoting an encompassing category that mediates and negotiates the values and languages of these two spheres (Fedele and Knibbe 2020, pp. 13–16). This allows institutional actors of secular organizations to bring eco-spiritual references into "what may be called the new, global 'knowledge class'" (Knoblauch 2014, p. 98). In Switzerland, some authors, inspired by Joanna Macy's work, are mediators between eco-spirituality and eco-activism and participate, on the one hand, in blurring the categories between religion and the secular while, on the other hand, reinforcing the religion and ecology nexus.

As in Macy's work, gendered dimensions are contained in the practices and discourses of the eco-spiritual activists we met, although they do not publicly elaborate on them. For instance, key speakers of the transition movement often explained how they tried to articulate an "internal transition" with an "external" one, thus advocating for a radical transformation of the self, while in return also affecting societal structures, moral and political norms, and means of production and consumption. The emphasis was then placed upon every individual self, a non-gendered self, promoting practices such as the "Work that reconnects", in which individuals, regardless of whether they were men or women, religious, spiritual or overtly non-religious, had to express their deep feelings regarding the degradation of the environment. Instead of elaborating on gender issues, key speakers of the transition network inserted them into larger philosophical considerations tending

toward universalization. The value given to self-expression or to emotion was not framed as a gendered, that is, cultural, issue here.

For instance, on the stage of one of the main theaters of Lausanne in March 2019, two authors, a Swiss philosopher we had already met and interviewed and a French author and biologist, were invited for a public conversation. The French author writing about what he calls "collapse", Pablo Servigne, started crying while detailing a video that struck him of an orangutan trying to resist bulldozers razing its habitat on the Island of Borneo. During that evening, he also asked the audience, 300 persons, including us, to undertake a brief exercise inspired by Macy's "Work that Reconnects": we had to greet our neighbor, look him/her in the eyes, and for two minutes describe in detail how we felt every time we thought about the loss of biodiversity worldwide. The other person had to remain silent, but be receptive and empathetic toward our words and feelings. Then, we exchanged roles for another round of two minutes. During that evening, Pablo Servigne also gave a description of the enchantment of the "Web of Life":

> The Web of Life has not unwoven for nearly 3.8 billion years, billions of years that [have spawned] the first bacteria, which are our ancestors. It is more than a string, it has woven a web, abounding, bushy and multicolored for billions of years. We are interdependent. We all have here strong interdependency with other living organisms outside this room. We are more than this room, and this radical interdependency is magnificent, she is moving and touching us. Additionally, in fact, it leads us toward something that is beyond us. That we even have a hard time imagining. It is something similar to sacred, what the Anglophones call "Wholeness", the feeling of unity with the Whole. (Our translation)

These observations and the quote illustrate how the key speakers valued and generalized specific forms of gendered attitudes, such as emotionality or showing that men can cry, yet without necessarily addressing overtly the gender issues at stake. They perform an inversion of gendered stereotypes through the identification with an animal, thereby conveying the idea as being beyond mere gender issues. Some of these key figures, though they did not mention it during conferences or workshops, did write about gender complementarity or about a so-called feminine side of the ecological transition (Egger 2017, p. 45). By relying on an apprehension of nature as a place of enchantment, or of "wholeness", these key figures were adopting a universalizing perspective. Just a few months later, a free film[5] projection was promoted by the same network on a popular cultural beach in Geneva. The movie documented the story of a social initiative in which adolescents living in suburbs were encouraged to disconnect from digital media in order to communicate with trees. Before and after the much-appreciated movie, a discussion took place with wood scientist Ernst Zürcher who featured in the film. The audience asked numerous questions about trees, their capacity to produce water and regulate urban heat, their relation to earth and to humans. The discussion contained a number of organicist metaphors, such as considering trees as organisms and forests as organs of the earth. The claimed aim of the organizers of the event was to raise awareness about the value of trees, which took different forms. While Zürcher explained complex processes performed by trees, such as photosynthesis, in scientific terms, he often used the word "cosmic" and followed others in referring to trees as male or female. When we asked him at the end of the event to elaborate further, he immediately pointed out that to consider trees male or female was indeed scientifically inaccurate. He added that speaking of male and female trees was, however, a way to communicate with people at what he called a "spiritual level". Here, it appeared that communicating at a spiritual level with urban dwellers meant attributing human gendered features to nature.

Within this network, gender was neutralized, while interdependency in the Anthropocene was universalized. Indeed, through the claims of holism, and by stating that every living organism was part of a "Whole", some elements of criticisms were side-stepped in the public communications. The rhetoric of self was inscribed into a wider view of interdependency between humans and non-humans, thereby marginalizing a critique based on

gender. In that regard, the use of organic metaphors when relating to nature, as Timothy Morton argues, neither encourages ecological views nor fosters subversive gender roles and models. For Timothy Morton, "Organicism is not ecological. In organic form the whole is greater than the sum of its parts [ . . . ] The teleology implicit in this chiasmus is hostile to inassimilable difference", inasmuch as "organicism polices the sprawling, tangled, queer mesh by naturalizing sexual differences." (Morton 2010, p. 278) This remark leads us to the last profile we found in the field, a profile revolving around a perception and conception of nature as uncanny and challenging toward a dominant, binary, gendered moral order.

*5.3. Neo-Shamanism: Queering Natureculture*

It is among some neo-shamanic practitioners that we found the idea of a non-organic and "queer mesh"; that is, what Morton describes as "a non-totalizable, open-ended concatenation of interrelations that blur and confound boundaries at practically any level: between species, between the living and the non-living, between organism and environment" (Morton 2010, p. 275). We shall focus on one practitioner in particular who integrated views on naturecultural patchworks in an explicit way by encouraging not only the sensing of spirits to connect with but also to discover the animal reflecting one's self. In the realm of neo-shamanic practitioners, we met mostly artists who had been initiated into neo-shamanism many years ago and even trained in Michael Harner's *Foundation for Shamanic Studies*. Those active in the ecological realm adopt sentient dimensions to frame ecological or nature-related issues. They offer what they call "forest baths", "vision quests", or "drum circles in the forest" (all expressions we found on leaflets or websites), for instance, and participate in ecological festivals. The practitioner we concentrate on, an artist photographer and journalist, was offering "queer yoga" classes in Geneva. In the long exchange we had with her, she elaborated on the way she thinks of nature and what "queer" means for her:

> We can make a direct link between neo-shamanism and queer. This link is situated in the relation to magic, in the relation to movement which is not fixed, not written, not stopped, not defined and in the porosity of the real, the porosity between the worlds . . . the relation to the spirits. To think that there are entities. There we enter into spirituality. To think that there is something other than just me and to think that there are energies of the entities of the gods [and] goddesses, it does not matter, we will say the word "spirit", if we imagine that there are the spirits and the human. Additionally, that we imagine that it's not separated and that it's possible to dialogue and to trade. Additionally, therefore a space where human and nature are not dissociated, where . . . human is nature. (Fieldwork interview, Lausanne, April 2019)

We can clearly recognize in this quote how the self is no longer at the center of the worldview or cosmology; on the contrary, there appears to be a hybrid whole—close to what Morton would call a mesh. According to Morton's notion of "dark ecology" (Morton 2018), when a deep connection with the earth develops, a sense of solitude appears. This solitude comes from the fact that humans realize that today, "contemporary capitalism and consumerism cover the whole Earth and reach deep into the forms of the living" and that "we actively and passively destroy life-forms inhabiting and constituting the bio-sphere, in Earth's sixth mass extinction event" (Morton 2010, p. 273). The neo-shamanic practitioner we quoted did indeed not favor a harmonious or colorful view of nature. The photographs she exhibited evoke a sense of mysticism, of cold fairyism with smoothed contrasts between soil, trees and sky (Figure 1).

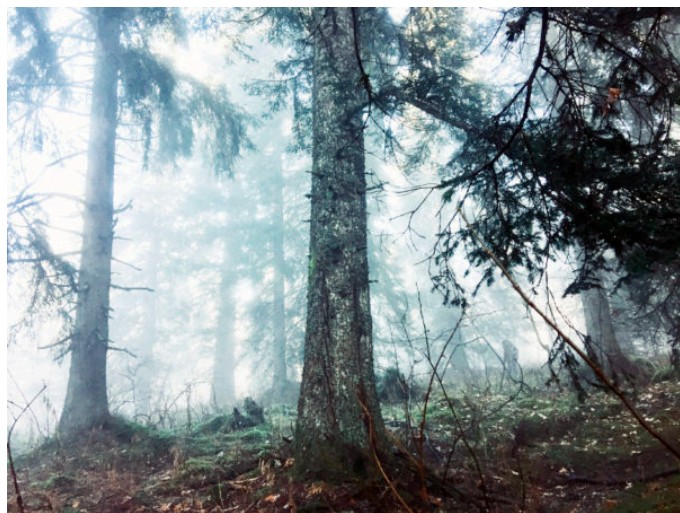

**Figure 1.** Photo from https://www.ceuxdici.ch/jeux-de-la-balle/ (reprinted with permission from Carine Roth (2020) accessed on 3 November 2021).

To actually be inside a mesh and talk about it puts her necessarily in an impossible position that is then reflected in the convoluted language she uses. In the rituals this shamanic practitioner proposes, she does not avoid puzzling and somewhat worrying encounters. Her non-conformist look in terms of clothes and gendered aesthetic contributed to this. Through her shamanic as well as daily practices and through her discourse about "spirituality" and "queer", this informant constantly performed a blurring of common categories. We conclude that she enacted naturecultural views in the sense pinpointed by Donna Haraway (2003).

## 6. The Emergence of Critical Stances at the Hubs of Eco-Spirituality

These three ways of articulating notions of self, ecological awareness, and gender constitute a continuum from an essentialist pole with polarizing and harmonious gender energies to the progressive disappearance of the gendered self and, finally, of the core self. These three positions leave little space for participants to frame a discussion of the power issues at stake regarding how gender and the ecological crisis relate to each other, as Christine Bauhardt does, for instance (Bauhardt 2013). Nevertheless, given the territorial proximity of these actors, some situations arise when such a space is created. One such situation occurred in July 2018 when Starhawk, the famous Californian feminist eco-activist and long-standing promoter of goddess spirituality,[6] was invited for a two-day workshop in Lausanne. The event was organized by a coalition formed by a local socio-cultural activity center, a Christian NGO, university chaplains, a pedagogic farm promoting permaculture, and an association of the degrowth movement. A high number of persons from the New Age and holistic milieu, the transition network, and the quoted neo-shamanic practitioner attended the event, making for a very heterogeneous audience. Women of all ages formed the majority, yet various male key figures of the inner circle of the transition movement were also present. In our fieldnotes we wrote:

> It is a Friday evening when we attend together with hundreds of interested people of all ages a public conference on ecofeminism featuring the key speaker Starhawk. The attending crowd has quickly filled the 200 seats in the space rented for the occasion near the main railway station, and more chairs need to be found to seat everybody. During half of the conference, an attentive and silent audience listens to Starhawk explaining her alternative archeological and mythological views on a pre-Christian matrifocal society. She argues that ancient Caucasian societies were "focused on the power of nurturing, of nourishment, of bringing life into the world of life-itself, and that the sacred [ . . . ] was embodied, imminent,

in life-itself, in nature itself, in the natural world". We are seated next to young women artists from the area who are very excited about meeting Starhawk in person as well as older scientists known from the university. Starhawk sums up her life-long posture upon spirituality and gender issues:

"You know, if we only see the divine and the sacred in male form, then it becomes very hard again as a woman, or as a gender-fluid person, or person who does not fall into these nice and neat binary categories, to feel that sacred connection to your own body and to your own life. To feel that you are an inherent carrier of value and that it also becomes difficult in society again to hold that we must value all of us. All of our images of who is really sacred and divine again only reflect one gender or one race or one color or one way of being."

She then introduced the audience to the alternative agronomy of permaculture, offering some basic organic farming principles and referencing such science as pedology—the study of the rhizosphere and the formation of arable soils. The audience raised several questions and praised the speaker and the organizers.

The particularity of Starhawk with regard to the actors quoted so far is that she explicitly elaborated on a feminist posture. She articulated a critique of the dominant patriarchal system with environmentalism and identified as a woman:

[It is] very important as women that we also look at the impact of the environment on us as women, and on how environmental degradation impacts women around the world, in ways that I think are even more extreme sometimes than how they impact men. Because, again, of this unequal structure of power. So if you have a society or you have a culture where women have less power, and less power in the home and less money and less resources, and less access to knowledge and education, you add to that environmental degradation. Then, it is often the women who go hungry to feed their family. It is the women who end up walking miles and miles and hours and hours each day to collect wood when the supplies around the home are used up, or to get water when there is no access to clean water. Therefore, we have to look at these issues, together. (Lausanne Fieldwork notes July 2018)

The theme of care and the conception of nature as a place of re-enchantment are clearly central to Starhawk, and mythological, ethical, spiritual, and scientific framings are easily entangled in her discourse. Although Starhawk emphasized idealized feminine values, a prehistoric matrifocal past or even the counter-cultural figure of the witch, she explicitly mentioned that she addressed women as much as "gender-fluid persons" or "persons who do not fall into these nice and neat binary categories", a statement loudly applauded in the room. The attendance of Starhawk, but also the popularity of this event, shed light upon the growing—in size, in visibility and in cross-references—number of feminist critics within eco-spiritual activism. In our case, Starhawk's conference brought some gender trouble into the local eco-spiritual milieu that was largely composed, as indicated, of a variety of actors promoting different views and practices with regard to ecology, gender, and spirituality. Starhawk's counter-cultural voice professed subversive considerations regarding gender identities as well as the nature/culture dichotomy. Starhawk indeed operates a subversion of gender attributes while still using gendered rhetoric, yet in a self-reflexive manner. Her speech entailed a common call to global ecofeminist and queer ecologists to have their voices heard and to value multiple codes in expressing gender and sexual orientation, especially when mediating critics related to the depletion of local biotopes.

Starting with Starhawk's conference, we observed that the articulation of the links between spirituality, ecology, and gender was becoming more and more popular. This articulation served communicative purposes within different social movements seemingly distant from the eco-spiritual network we studied. For instance, in June 2019, a historical mobilization of "women*"[7] took place all over the country. In the city of Lausanne, the women*'s strike started in an unsuspected manner: at midnight a walk was organized

resonating with a witches' "Walpurgis" night. It gathered hundreds of women* around a bonfire where they ritually burned bras. In 2020, the event was repeated, but the different events were scattered across several places due to COVID-19 restrictions. A few days after the strike, in the back street of a gentrified neighborhood of Lausanne, one signboard remained for everyone to see on the walls of an autonomous socio-cultural center. It stated (in French) that "We are not fighting for a white, masculine, and heteronormative nature/we are a feminist, black, sacred, living nature defending itself". The iconography is sober, featuring a pine tree, weeds, and even a dead tree, all of it encompassed by the generic category of nature (Figure 2). There is a "dark ecology" present in this signboard as nature is also represented by its misfits and unaesthetic wild weeds.

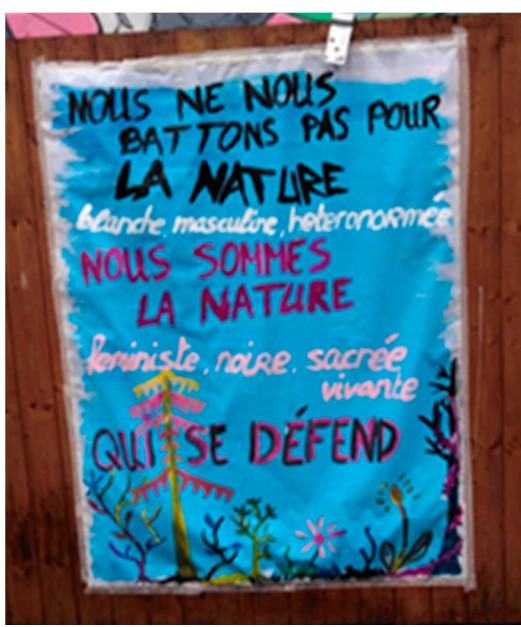

**Figure 2.** Photo by Alexandre Grandjean, Lausanne, 16 June 2020.

The poster in Lausanne clearly contrasts a "masculine" ethos with a wider "feminist" ethos, thereby encouraging women to go beyond a view of gender equality and to engage in intersectional militancy. By qualifying Nature as sacred, this poster also hints at the idea that "gendered ritual contact with the otherworld does in some circumstances empower women" (Greenwood 2000, p. 148). Furthermore, there is not only a view of gender polarity but also clearly an ideal of the feminine being more powerful than the masculine. Such an ecofeminist view does contain a form of the new "subtle spirituality" that is locally still seeking its social, cultural, political and discursive location (Becci et al. 2021). Interestingly, the different views that seek to re-enchant Nature, but also provide a critical stance, are also a way of re-enchanting the militant self and strengthening the political ability of social movements to engage in intersectional struggles where climate imperilment, social justice, and gender equalities go hand-in-hand. This small example also illustrates possibilities to go beyond the essentialist or queer endeavor we have presented in this article in order to form new militant coalitions. It is, however, too soon to draw conclusions on the social dynamics and impacts of such intersectional views on nature.

## 7. Conclusions: Gender, Nature, Religion, and the Anthropocene

Our findings point to a variety of ways by which a nature referred to as "sacred" is gendered, ending either in essentialist views or in queer perspectives. In this article, we have considered the notion of nature as a surface on which to project views and representations about one's self, others, and the environment. Yet, as we have carefully shown, if it is possible to delineate three distinct postures relating to holistic spiritualities, inner transition and neo-shamanism, we have also shown how these different postures tend not

to be isolated. On the contrary, as shown by the ethnographic vignette of Starhawk's conference presentation in Lausanne in 2018, these eco-activists frequently encounter each other during public events. Eventually, with regard to what we have termed a "spiritualization" of ecology (Becci and Grandjean 2021), these key figures are important as they promote practices and worldviews in which the self, gender and nature entangle in various ways. Yet, in the public domain, these eco-spiritual registers tend to be popularized; that is, they are adapted to the communication codes of secular culture and even contemporary political movements, such as feminism.

In summary, beyond the great variety of forms of spirituality that appear on the ground at first sight, with disparate references and the incessant assertions of actors aiming to differentiate themselves from each other, an anthropological perspective allows us to distinguish a series of analogies that construct a new vision of the social order. In the observations we made, thinking in analogies constitutes a holistic thinking that tends toward universalization and a neutralization of cultural differences. The recurring principles are the holistic beliefs in harmony and the values of gratitude, non-violence, fluidity, and economic sobriety. Through our empirical observations and analysis of the eco-spiritual milieu in Switzerland, we found that the change in representations of nature, and the introduction of new religious expressions also redraw gender norms and stereotypes. We have shown that universalistic and planetary imaginaries such as the "Web of Life" can be contextualized and related to individuals' gendered selves. Such a contextualization is necessary to consider the so-called "Anthropocene" in sociological terms and identify, as Donna Haraway (2003) suggests, the location of social, political, and economic capital in ecological action.

**Author Contributions:** Conceptualization, I.B. and A.G.; funding acquisition, I.B.; investigation, I.B. and A.G.; methodology, I.B. and A.G.; project administration, I.B.; supervision, I.B.; writing—original draft, I.B. and A.G.; writing—review and editing, I.B. and A.G. All authors have read and agreed to the published version of the manuscript.

**Funding:** Swiss National Science Foundation. Project 169823.

**Institutional Review Board Statement:** Ethical review and approval were waived for this study, as no sensitive data were used and the observed events were public.

**Informed Consent Statement:** Informed consent was obtained from all subjects involved in the study.

**Acknowledgments:** The authors thank the editors of this issue for their insightful comments, advice, and critiques of previous versions of the text.

**Conflicts of Interest:** The authors declare no conflict of interest.

## Notes

[1]　These notions often have an ambivalent status as they are used by scientists as well as by practitioners of contemporary spirituality (see for instance Narby (1995), p. 68 or Zürcher (2016), p. 41, for "cosmic" (serpent, tree), and Glowczewksi and Pruvost (2021) for the notion of "healing energies").

[2]　For instance, https://painpourleprochain.ch/transition.interieure/ (accessed on 21 November 2021).

[3]　For further insights into these analytical research categories the reader may refer to the different contributions made in the edited book *Secular Societies, Spiritual Selves?* by Fedele and Knibbe (2020).

[4]　The book which contributed enormously to the introduction of international ecofeminist texts to a Francophone context was Emilie Hache's collection *Reclaim* (Hache 2016), published in 2016.

[5]　The movie shown was entitled "Connectés" (Schinasi 2018). For more information on that day, see https://aux-arbres.com/les-solutions/ (accessed on 21 November 2021).

[6]　See Salomonsen (2002) for a close description of this network. Starhawk's political writing on eco-activism had recently been translated into French (Starhawk 2015), though her writing on religious and gender creativity, such as the bestseller *The Spiral Dance* (Starhawk 1979), still remains unavailable in French.

[7]　For this event celebrating a historical women's strike in 1991, numerous debates occurred around the presence of an official asterisk to mark inclusion of transgender and non-binary persons.

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
