# Peer review of "Is Sacred Nature Gendered or Queer? Insights from a Study on Eco-Spiritual Activism in Switzerland"

_religions, doi:10.3390/rel13010023_

Round 1
Reviewer 1 Report
I found the first part of the essay less easy to read. The style of the authors improves as the article progresses. Perhaps it was because I had not read as much on eco-spirituality. I very much appreciate the three categories of persons involved in the eco-spiritual. The connection to gender is good and helpful ... but was confusing as the authors laid out the first section.
I appreciated the author's more ethnographic approach to the research as they interviewed people and attended events in Geneva, Switzerland! Very helpful to see eco-spirituality operating in a community.
Author Response
Thanks for the comments. We change considerabIy the first part of the essay to make it clearer for the readers.
We put a stronger emphasis on which perspectives we studied to distinguish more neatly the emic discourses from our analysis.
Reviewer 2 Report
The authors rightly point out the growing role of the spiritualisation of ecology. The literature more and more often indicates the need to take into account the spiritual/religious dimension of nature. Showing the dimension affects human relations with nature, supplementing legal, aesthetic, and rational arguments with arguments referring to the sacred. The authors try to combine three elements in an interesting way: spirituality, ecology, and gender. The idea itself is interesting. Its implementation, however, encounters certain difficulties. It is based on very illusory arguments. For example, someone identified Egyptian goddess Isis in nature through feeling (verse 227). People sense male and female energies in a similar way. Criticizing "disenchantment of the World" led by western modernity (verse 261) leads to extreme subjectivity and esoterism, which does not even bear the hallmarks of scientificity. All this leads to the conclusion that the authors use extremely subjective methods, focus on the opinions of selected people (verse 391), and then generalize the conclusions implied by these opinions.
Overall, I find the manuscript innovative and interesting. However, I suggest that the text be thoroughly edited, pointing out that the authors talk about people's beliefs, not facts. The point is that when presenting their research, the authors emphasize that they report the beliefs of respondents to their research.
Author Response
Thanks for the comments. We have changed several parts of the text, in particular the first part of the essay to make it clearer for the readers how we proceeded. We put a stronger emphasis on which perspectives we studied to distinguish more neatly the emic discourses from our analysis.
We have detailed our methodology in order that it becomes evident that the arguments that are found as "illusory" are not taken as facts but as people's beliefs we report about.
Reviewer 3 Report
The key figures interviewed are mostly male. Is this a reflection of the milieu? Are there no female key figures?
page 4, line 158-164: the section as a whole defines concepts, but here 'emic' notions of masculine and feminine are discussed. Or are these etic notions, that are later broken through the introduction of the notion of queer? Or emic notions of masculinity and feminity (which are indeed often quite binary and essentialism in these circles)? In any case, the discussion on the Masculine and Feminine as well as later on concerning the introduction of the notion of queer is not clearly flagged as either part of the conceptual framework or 'emic' notions the author found through their research. If they are both (which is certainly possible), this could be mentioned as well.
172-174: this could be rephrased as a question, it seems to me. It's a bit hard to place this statement otherwise.
332 onward: the translation of the quote contains a few errors and typo's.
366-375 I have a hard time following this paragraph. The first sentence is already confusing: "We argue that the use of organic metaphors when relating to nature leave little space 366 for elaborating on gender issues" Space for whom to elaborate? And what does 'elaborating on gender issues' actually mean? Further on, when speaking of 'inassimiliable differences' it seems they are implying that the people described do not subscribe to essentialist views of binary gender, but I am not sure if this is what the authors mean. The in 373, the phrase 'elements of critic' is used. I assume this should be 'critique', but whose critique and what is the critique? Is this the critique of essentialising gender? In short, I have to assume what this is about, but I'm not sure that I am reading this right.
The conclusion contains some new information. This might be a problem depending on your view of what a conclusion should be. An easy solution would be to call this a discussion.
Overall, I can follow the structure of the paper and the argument, and it certainly describes very interesting trends. I do wonder, however, whether aside from the harmonious coming together of the three different positions on gender in events such as the conference around Starhawk, clashes also occur. It would make the paper stronger if they could say something about this.
Author Response
Thanks for the very useful comments.
In a previous text we analyse why the key figures interviewed are mostly male. The reference is in the essay.
We have clarified more thouroughly which notions are emic or etic.
We have rephrased the unclear sentences on the lines 172-174, 332 onward, and paragraphe 366-375.
We changed the title "conclusion" into "discussion".